# Diagnostic Potential of Endometrial Cancer DNA from Pipelle, Pap-Brush, and Swab Sampling

**DOI:** 10.3390/cancers15133522

**Published:** 2023-07-06

**Authors:** Yinan Wang, Hui Du, Wenkui Dai, Cuijun Bao, Xi Zhang, Yan Hu, Zhiyu Xie, Xin Zhao, Changzhong Li, Wenyong Zhang, Ruifang Wu

**Affiliations:** 1Department of Obstetrics and Gynecology, Peking University Shenzhen Hospital, Shenzhen 518036, China; yinanwang@pkuszh.com (Y.W.); duhui_107108@163.com (H.D.);; 2School of Medicine, Southern University of Science and Technology, 1088 Xueyuan Avenue, Shenzhen 518055, China; zhangwy@sustech.edu.cn; 3Institute of Obstetrics and Gynecology, Shenzhen PKU-HKUST Medical Center, Shenzhen 518036, China; 4Shenzhen Key Laboratory on Technology for Early Diagnosis of Major Gynecologic Diseases, Shenzhen 518036, China; 5Department of Clinical Medicine, Xi’an Jiaotong University, Xi’an 710049, China; 6China National GeneBank, BGI-Shenzhen, Shenzhen 518116, China; zhaoxin@genomics.cn

**Keywords:** endometrial cancer, early diagnosis, endometrial specimens, minimally invasive methods, targeted NGS panel

## Abstract

**Simple Summary:**

Endometrial cancer is a significant and growing health concern worldwide. In this study, researchers aimed to find a safe and practical way of detecting early signs of endometrial cancer, which is crucial for effective treatment and improved patient outcomes. The researchers designed a panel targeting specific genes related to endometrial cancer and tested it using samples from 38 endometrial cancer patients and 208 women with risk factors. The results showed that the panel performed well, producing high-quality data and detecting genetic mutations with high sensitivity. The best sample type for detecting mutations was endometrial biopsy using the Pipelle aspirator, which had the best consistency with surgical tumor specimens. The findings suggest that this targeted panel sequencing method combined with ultra-deep sequencing is a promising tool for the early detection of endometrial cancer and could have significant clinical implications for patients.

**Abstract:**

Endometrial cancer (EC) is a major gynecological malignancy with rising morbidity and mortality worldwide. The aim of this study was to explore a safe and readily available sample and a sensitive and effective detection method and its biomarkers for early diagnosis of EC, which is critical for patient prognosis. This study designed a panel targeting variants for EC-related genes, assessed its technical performance by comparing it with whole-exon sequencing, and explored the diagnostic potential of endometrial biopsies using the Pipelle aspirator, cervical samples using the Pap brush, and vaginal specimens using the swab from 38 EC patients and 208 women with risk factors for EC by applying targeted panel sequencing (TPS). TPS produced high-quality data (Q30 > 85% and mapping ratios > 99.35%) and was found to have strong consistency with whole-exome sequencing (WES) in detecting pathogenic mutations (92.11%), calculating homologous recombination deficiency (HRD) scores (r = 0.65), and assessing the microsatellite instability (MSI) status of EC (100%). The sensitivity of TPS in detection of EC is slightly better than that of WES (86.84% vs. 84.21%). Of the three types of samples detected using TPS, endometrial biopsy using the Pipelle aspirator had the highest sensitivity in detection of pathogenic mutations (81.87%) and the best consistency with surgical tumor specimens in MSI (85.16%). About 84% of EC patients contained pathogenic mutations in *PIK3CA, PTEN, TP53, ARID1A, CTNNB1, KRAS,* and *MTOR*, suggesting that this small gene set can achieve an excellent pathogenic mutation detection rate in Chinese EC patients. The custom panel combined with ultra-deep sequencing serves as a sensitive method for detecting genetic lesions from endometrial biopsy using the Pipelle aspirator.

## 1. Introduction

Endometrial cancer (EC) is one of the three major gynecological malignancies, and its morbidity and mortality are rising worldwide [1]. The five-year overall survival rate for EC patients diagnosed in the early stage is more than 81%. However, the five-year survival rate for 33% of patients with late-stage EC at diagnosis plummets to 15% [2]. Early diagnosis of EC is critical for patient prognosis, but there is still no accurate and effective method.

Currently, EC is diagnosed and confirmed by histopathological analysis of endometrial biopsy samples obtained by curettage and hysteroscopy. Histopathological diagnosis is inconclusive in up to 30% of patients [3,4]. In addition, the sensitivity and specificity of positive signs (i.e., vaginal bleeding) and auxiliary examinations, such as cervical smear, transvaginal ultrasound scan, and tumor marker examination (e.g., protein cancer antigen 125), are insufficient [5]. Early diagnosis of EC in the potential risk population required an optimal technology that needed a reasonable cost with acceptable minimally invasive sampling methods [5]. The anatomical continuity of the uterine cavity with the cervix makes vaginal and cervical samples, along with endometrial samples, candidates for minimally invasive sampling [5,6]. Tumor cells shed from EC are carried into the endocervical canal or vagina [6]. The Pap brush or cervical–vaginal swab can capture these cells [6,7,8]. However, the number of tumor cells in vaginal and cervical samples is much lower than that of normal cells. An analytic technique was needed to identify mutant alleles among large quantities of wild-type alleles reliably. Recent advancements indicate that genomic analyses in minimally invasive sampling are an essential step toward a widely applicable methodology for the early detection of EC [9]. Kinde et al. used targeted sequencing to identify the same mutations in Papanicolaou (Pap) specimens in 100% of ECs (*n* = 24), laying a foundation for the diagnosis of EC by high-throughput sequencing [6]. Wang et al. applied an 18-gene panel to sequence Pap-brush samples from 382 women with EC and 714 without cancer. The sensitivity was 81%, and the specificity was 99% [8]. Reijnen et al. detected mutations in eight genes for self-collected vaginal specimens, cervical Pap-brush samples, and Pipelle endometrial biopsy from 59 EC patients and 31 controls. They found that the sensitivity of these three types of samples was 67, 78, and 96%, with a specificity of 97, 97, and 94%, respectively [7]. Genomics approaches have shown significant sensitivity and specificity for detecting EC in these samples. However, this approach calls for further assessment of the diagnostic accuracy of mutational analyses compared to traditional histopathological diagnosis to be introduced as EC diagnosis tests for women with risk factors/EC patients.

Thus, the aim of this study was to design a panel targeting known variants for EC-related genes for cost-effective and sufficient depth of sequencing and assess which Pipelle endometrial (PIP-E) biopsy, cervical Pap-brush (PAP-C) sample, and vaginal swab (SWAB-V) specimen is the optimal sample. We recruited 38 EC patients and 208 women with risk factors for EC to collect the above three samples and, when available, tumor and cancer-adjacent normal tissue samples. Then, we performed whole-exome sequencing (WES) on the tumor and matched normal samples for panel customization and targeted next-generation sequencing (NGS) on PIP-E, PAP-C, and SWAB-V for accuracy validation.

## 2. Materials and Methods

### 2.1. Patients and Sample Collection

In this prospective study, we recruited 40 suspected EC patients and excluded one uterine leiomyoma and one cervical adenocarcinoma patient according to postoperative pathology. Of the remaining 38 patients, the gynecologists collected 18 vaginal specimens using swabs, 29 cervical samples using Pap brushes, and 27 endometrial biopsies using Pipelle aspirators sequentially before surgery. The 38 EC patients each provided one surgery tumor sample for histopathological diagnosis and the other tumor and matched cancer-adjacent normal tissue sample for sequencing (Figure 1).

Moreover, we recruited 228 women with risk factors for EC and excluded 21 women with cervical intraepithelial neoplasia (CIN) based on the pathological findings. The remaining 208 women meet at least one of the following risk factors for EC: (i) irregular vaginal bleeding during non-pregnancy or delayed menopause (≥53-year-old); (ii) older than 45 years; (iii) older than 35 years and having one of the following: long-standing infertility; long-term estrogen stimulation; obesity; hypertension; diabetes; and family history of hereditary non-polyposis colorectal cancer, colorectal cancer, breast cancer, or EC. These 208 women provided 208 PIP-E, 208 PAP-C, and 198 SWAB-V samples, as shown in Figure 1 and Appendix A. They also provided uterine curettage samples from formal D&C under anesthesia for histopathological diagnosis.

We also gathered all patients’ epidemiological and clinical pathological information (age, BMI, histological subtype, menopausal state, endometrial thickness, etc.). The Peking University Shenzhen Hospital Ethics Committee approved this study (approval date 29 May 2015). All of the patients signed the informed permission papers. Following the National Health and Family Planning Commission of the People’s Republic of China’s instructions, we conducted all trials.

### 2.2. WES and Data Analysis

To provide a target gene list and verify the accuracy of the custom panel, surgical specimens, i.e., 38 tumors and matched cancer-adjacent normal tissue samples from the above EC patients, were used for WES. Fresh surgical samples (tumor and normal) were stored at −80 °C until DNA extraction. Specimen collection to DNA extraction took place over the course of a week. DNA extraction, library preparation, and sequencing were conducted by BGI (BGI, Shenzhen, China) as a commercial service. The detailed DNA extraction and library preparation protocol is available in the BGI technical support documents [10,11]. A MGIEasy Exome Capture V4 Probe Set was used for hybridization. The probe regions were 59 Mb, covering over 20,000 genes. WES was processed on the MGISeq-2000 platform with mean depth coverage of 750×, and the mode was set to PE100.

After obtaining raw read sequencing data, SOAPnuke [12] removed adapters and filtered low-quality reads. An average of 0.44 billion clean reads per sample were obtained. Sequencing quality was evaluated by FastQC v0.11.9 (https://www.bioinformatics.babraham.ac.uk/projects/fastqc/, accessed on 8 January 2020) and MultiQC v1.10 [13]. BWA [14] aligned clean reads to the human reference genome (GRCh37) with the default parameters. Then, the Genome Analysis Toolkit (GATK v 4.1.2.0) [15] was used to call somatic variants following the GATK Best Practices pipeline. The VariantFiltration module of GATK filtered variants with the following hard-filtering expressions “QD < 2.0”, “FS > 60.0”, “MQ < 30.0”, and “DP < 8.0” and BCFtools v1.10.1 (https://samtools.github.io/bcftools/, accessed on 17 December 2019) filtered variants with DP ≥ 2. Ensembl Variant Effect Predictor v93.7 (https://www.ensembl.org/info/docs/tools/vep/index.html, accessed on 11 October 2018) was utilized for variant annotation. Tumor mutational burden (TMB) was calculated as the number of non-synonymous somatic mutations per Mb of the target region. The microsatellite instability (MSI) was assessed using MSIsensor [16]. MSI is classified as stable (MSS, MSI score < 0.07), high (MSI-H, MSI score > 0.08), or uncertain (MSI-U, 0.07 ≤ MSI score ≤ 0.08). The homologous recombination deficiency (HRD) score was calculated by scarHRD [17], combining three indicators: heterozygous deletion (LOH), telomere allelic imbalance (TAI), and large segment migration (LST). HRD score, a continuous variable, was used as an index for subsequent analysis.

To obtain high-confidence pathogenic mutations, we filtered variants as follows: (1) variants with depth (DP) ≤ 10 were filtered out; (2) variants with allele depth ≤ 3 were filtered out; (3) variants with population frequencies > 1% were filtered out; (4) variants with the “common_variant” flag (allele frequency across at least one gnomAD subpopulation is >0.04%) were excluded; (5) variants annotated as pathogenic or likely pathogenic were retained for subsequent analysis. The population frequencies were obtained from the 1000 Genomes Project [18], Exome Aggregation Consortium East Asian dataset (ExAC_EAS, https://gnomad.broadinstitute.org/downloads#exac-variants, accessed on 8 July 2021), and gnomAD_EAS (https://gnomad.broadinstitute.org/, accessed on 27 May 2020).

### 2.3. Targeted NGS Panel Customization, Sequencing, and Analysis

Our custom hybrid capture panel covers coding exons and regions prone to mutations of 10,929 genes that had variants in the above WES results or had been reported in the literature of EC [6,7,8,19]. The list of targeted genes is in Appendix A. Oligonucleotide probes were designed by BGI. The probe regions were 4.96 Mb, and the average sequencing depth was 20,000×.

All the samples were profiled by targeted panel sequencing (TPS). Fresh surgical samples (tumor and normal) and PIP-E samples were stored at −80 °C until DNA extraction. PAP-C and SWAB-V samples were preserved in ThinPrep PreservCyt solution at 4 °C (Hologic, Marlborough, MA, USA). After delivery to the laboratory, PAP-C and SWAB-V samples were centrifuged for 5 min at 14,000× *g*, and the precipitates were used for DNA extraction. DNA extraction, library preparation, and targeted NGS were performed as described above. Raw sequencing data of EC patients were also processed like WES data. Sequencing data from women with risk factors for EC were identified as variants using the HaplotypeCaller module of GATK.

### 2.4. Statistical Analysis

Data normality was verified using the Shapiro–Wilk test in SPSS version 22 (IBM, Armonk, NY, USA). Pairwise comparisons were performed by the Wilcoxon matched-pairs signed rank test. Multiple-group comparisons were performed using the Kruskal–Wallis test followed by Benjamini–Hochberg (BH) correction. The correlation was calculated with the Spearman rank test. Cohen’s kappa statistic (*K*) was used to assess the agreement between WES and TPS. R v4.0, GraphPad Prism 8 (GraphPad Software, La Jolla, CA, USA), Oviz-Bio platform (https://bio.oviz.org/demo-project/analyses/landscape, accessed on 2 July 2020) [20], and Adobe Illustrator (v2021, Adobe, San Jose, CA, USA) were used for Statistics and Visualization. *p* < 0.05 or false discovery rate (FDR) indicated statistical significance.

## 3. Results

### 3.1. Cohort Characteristics of Study Participants

In total, 246 Chinese women participated in the study, including 38 patients with EC and 208 women with risk factors for EC (Figure 1). No patients overlapped between these two groups. The median age of the EC and potential risk groups was 56 years (range 33–65) and 45 years (range 23–75), respectively. The median BMI was 24 kg/m^2^ for the EC group and 23 kg/m^2^ for the potential risk group. EC included endometrioid adenocarcinoma (27, 71.05%), serous carcinoma (6, 15.79%), and other nonendometrioid carcinomas (5, 13.15%) in the EC group. Most EC patients were in menopause (25, 65.79%) and had vaginal bleeding (34, 89.47%).

In the potential risk group, ten women were detected to have precancerous lesions of EC, i.e., one atypical endometrial hyperplasia, eight endometrial hyperplasias, and one cystic endometrial atrophy. Moreover, 69 women had benign lesions, including chronic endometritis (*n* = 23), endometrial polyp (*n* = 4), adenomyosis (*n* = 1), and benign cystic glandular hyperplasia (*n* = 41). There were 50 women with unchanged endometrium (in the phase of proliferation and secretion). About 72% (149/208) of women had vaginal bleeding, and most (194, 93.27%) had an endometrial thickness of ≥4 mm. Sixteen, nine, and twenty-five women had hypertension, diabetes, and a family history of cancer, respectively (Table 1). 

### 3.2. Performance of the Custom Panel in Surgical Specimens

We first assessed the quality of WES and TPS data. After removing adaptor sequences and low-quality reads, an average of 1.14 billion clean reads per sample was obtained by TPS. The base Q30 values and the mapping ratios for all TPS samples were greater than 85% and 99.35%, respectively. Compared with WES, TPS had a higher rate of regions with coverage depth over 100× (Cov100×, 97.65% vs. 80.71%) due to its higher sequencing depths (Appendix A). From the above results, the data from TPS passed quality control checks.

To assess the performance of the TPS for detecting somatic mutations, we compared its results with those of WES and postoperative pathology reports (Figure 2a). All patients harbored detectable mutations either by TPS or WES. TPS and WES identified pathogenic mutations in 33 and 32 EC patients, respectively (Figure 2b). Compared with the clinical pathological results, the sensitivity of TPS and WES in detection of EC was 86.84% (33/38) and 84.21% (32/38). TPS and WES identified pathogenic mutations in 31 patients and none in four, indicating strong consistency between TPS and WES (92.11%, *K* = 0.68, *p* < 0.001). About 72.76% (219/301) of the pathogenic mutations detected by WES were also detected by TPS (Figure 2c and Appendix A).

Following the results of WES, 84.21% (32/38) of the patients with EC had pathogenic mutations. The most common genes with pathogenic mutations were *PIK3CA* (53%, 17/32), *PTEN* (38%, 12/32), *TP53* (19%, 6/32), ARID1A (16%, 5/32), *ABCA12* (12%, 4/32), *CTNNB1* (12%), and *KRAS* (12%). About 76% (29/38) of EC patients contained pathogenic mutations in *PIK3CA*, *PTEN*, *TP53*, and *ARID1A*. The addition of each of the three genes, *CTNNB1*, *KRAS*, and *MTOR*, increased by 2.63% in the pathogenic mutation detection rate (Figure 2a and Appendix A). TPS also detected pathogenic mutations in the above seven genes in 32 patients (Figure 2a and Appendix A). These seven genes can be combined to form a gene panel with a good pathogenic mutation detection rate in Chinese EC patients.

We analyzed TMB, MSI, and HRD to explore clinical utility based on TPS and WES data. The TMB of TPS and WES showed significant correlations (Spearman test, r = 0.87, *p* < 0.001), whereas the TMB of TPS was lower than that of WES (Wilcoxon matched-pairs test, *p* < 0.001). The HRD of TPS and WES showed no differences (Wilcoxon matched-pairs test, *p* = 0.10; Spearman test, r = 0.65, *p* < 0.001). All MSI status recognized by TPS was consistent with those by WES (Figure 2d and Appendix A). Taken together, our custom TPS had good consistency with WES in detecting pathogenic mutations, calculating HRD score, and assessing the MSI status of EC.

### 3.3. TPS Data Quality of Endometrial, Cervical, and Vaginal Samples 

We assessed the TPS data quality of 215 endometrial biopsies, 228 cervical samples, and 206 vaginal specimens from EC and potential risk groups. The numbers of clean reads obtained from surgical specimens, endometrial biopsies, cervical samples, and vaginal specimens were not different (*p* = 0.82; Figure 3a). About 1.17 billion clean reads were obtained per endometrial, cervical, and vaginal sample. The Q30 of sequencing data, as well as the Cov100×, did not differ between these types of samples (Q30: *p* = 0.21; Cov100×: *p* = 0.77; Figure 3b,c). The mapping ratios of endometrial biopsies and surgical specimens had no difference (FDR = 0.09) and were higher than those of cervical and vaginal samples (FDR < 0.0001; Figure 3d). In conclusion, the data from endometrial, cervical, and vaginal samples were qualified, and the data from endometrial biopsies were of better quality.

### 3.4. Early Detection Effectiveness of Endometrial, Cervical, and Vaginal Samples for EC Patients and Women with Risk Factors

Of the 38 EC patients, TPS data of endometrial biopsies, cervical samples, and vaginal specimens were available for 27, 29, and 18 women, respectively (Figure 1). We compared these three types of samples with their matched surgical tumor specimens to assess the diagnostic value of these three samples (Figure 4a). After stringent filtering, TPS detected somatic mutations in 100% of endometrial, 96.55% of cervical, and 94.44% of vaginal samples. Pathogenic mutations were identified in 81.48% (22/27) of endometrial, 55.17% (16/29) of cervical, and 44.44% (8/18) of vaginal samples, respectively (Figure 4b and Appendix A). Thus, the sensitivity in detection of mutations of EC of endometrial biopsy was the best. About 78.80% (145/184), 53.92% (117/217), and 42.86% (48/112) of the pathogenic mutations detected in surgical specimens were also detected in endometrial, cervical, and vaginal samples, respectively (Figure 4c and Appendix A). The pathogenic mutation detection rates of the gene set comprising *PIK3CA*, *PTEN*, *TP53*, *ARID1A*, *CTNNB1*, *KRAS*, and *MTOR* in PIP-E, PAP-C, and SWAB-V samples were 100% (22/22), 93.75% (15/16), and 87.5% (7/8) in patients with detectable pathogenic mutations, respectively (Appendix A and Figure 4). The endometrial biopsy had the best consistency of pathogenic mutation calls with the surgical tumor specimen (PIP-E: *K* = 0.71, PAP-C: *K* = 0.58, SWAB-V: *K* = 0.70). Otherwise, the TMB of WES from the surgical specimen was different from that of TPS from endometrial biopsy (*p* < 0.0001) and not different from that of the cervical (*p* = 0.88) and vaginal sample (*p* = 0.24; Figure 4d). Precisely the opposite was observed in HRD (Figure 4e). There was no difference between the HRD of TPS from endometrial biopsy and the HRD of WES from the surgical specimen (*p* = 0.56), but the correlation was poor between them (Spearman test, r = 0.09, *p* = 0.32). Moreover, the best agreement with the MSI of WES (or TPS) was obtained from endometrial biopsy (85.16%; Figure 4f and Appendix A). Overall, the consistency between the endometrial biopsy and the surgical tumor specimen is the best.

Of the 208 women with risk factors for EC, TPS data of endometrial biopsies, cervical samples, and vaginal specimens were available for 188, 199, and 188 women, respectively (Figure 1). Pathogenic mutations were identified in 70, 66, and 64 women using endometrial, cervical, and vaginal samples, respectively (Appendix A). Based on the pathology report, women could be divided into three subgroups: hyperplasia group (including women with atypical endometrial hyperplasia and endometrial hyperplasias); benign group (including women with chronic endometritis, endometrial polyp, benign cystic glandular hyperplasia, and adenomyosis); and unchanged group (including women in in the phase of proliferation and secretion). We detected pathogenic mutations in 77.78% (7/9) of women in the hyperplasia group, 33.33% (23/69) of women in the benign group, and 42.00% (21/50) of women in the unchanged group (Figure 5 and Appendix A). Pathogenic mutations occur more frequently in the hyperplasia group, although this might also be due to potential errors caused by the small sample size. Otherwise, considering gene mutation is an early but insufficient event in tumorigenesis, women with pathogenic mutations in the potential risk group may have a higher risk of EC and require intervention or follow-up.

## 4. Discussion

Early diagnosis of EC is challenged due to the lack of a safe and readily available sample, sensitive and effective biomarkers, and detection methods. Gene abnormalities are essential biomarkers in EC [19]. Detection of such abnormalities in specimens collected using a Tao brush [8] or uterine lavage [21,22] rather than in tumor tissue has attracted significant attention because of its convenience, easy operation, and low cost. EC diagnosis and prevention can be improved by molecular analyses of specimens obtained using minimally invasive sampling methods [19]. 

In this prospective single-center study, we assessed the technical performance of our custom panel in detecting mutations by comparing with WES and the diagnostic potential of endometrial biopsy using the Pipelle aspirator, cervical sample using the Pap brush, and vaginal specimen using swabs by applying TPS to them and their matching surgical tumor specimens from 38 patients with EC. TPS produced high-quality data at high depth and was found to have strong consistency with WES in detecting pathogenic mutations, calculating HRD scores, and assessing the MSI status of EC. The sensitivity of TPS is slightly better than WES (86.84% vs. 84.21%) due to its ultra-deep sequencing. Of the three types of samples detected using TPS, endometrial biopsy using the Pipelle aspirator had the highest sensitivity in detection of pathogenic mutations (81.87%) and the best consistency with surgical tumor specimens in MSI analyses.

Moreover, we found that seven genes, i.e., *PIK3CA*, *PTEN*, *TP53*, *ARID1A*, *CTNNB1*, *KRAS*, and *MTOR*, can achieve an excellent pathogenic mutation detection rate in Chinese EC patients, regardless of sample type. This gene set contained well-known cancer driver genes *ARID1A*, *PTEN*, *PIK3CA*, and *CTNNB1* [23]. *PTEN*, *ARID1A*, *CTNNB1*, *PIK3CA*, and *KRAS* mutations commonly occur in type I EC [24,25], whereas *TP53* mutations are frequent in type II (90%) and associated with serous carcinoma and endometrial intraepithelial carcinoma [26,27]. Some gene mutations are associated with morphological changes in the endometrium. *PTEN*, *PIK3CA*, and *KRAS* mutations are known to be involved in endometrial hyperplasia, the precursors of EC [28,29]. Moreover, *CTNNB1* mutations were associated with squamous differentiation and mucinous differentiation absence [30]. This small gene set, combined with lower genital tract sampling, i.e., vaginal swabs or cervical brushes, is cheaper and easier to operate. It is a viable option for large-scale applications.

This study provides a sensitive assay and optimal sample for EC detection. Still, the specificity of this approach needs to be evaluated with a large number of healthy controls before it can be applied to clinical settings. However, it is encouraging that few pathogenic mutations have been detected in healthy controls so far in any of the evaluations. In the “PapGene” test, no mutations were detected in the 14 Pap specimens from healthy controls using targeted sequencing [6]. In the “PapSEEK” test, the specificity of the endocervical sample with Pap brush and the endometrial sample with a Tao brush was 99% (704/714) and 100% (125/125) [8]. Only one healthy woman (1/385) had one somatic mutation in her cervical sample in the Peremiquel-Trillas et al. study [31]. Reijnen et al. found that the specificity of Pipelle endometrial biopsies, Pap-brush samples, and self-samples were 94, 97, and 97% [7].

In addition, we have carried out a valuable exploration of EC early diagnosis for women with risk factors. We found that pathogenic mutations were more likely to be detected in women with endometrial hyperplasia, which was a precancerous lesion of EC. Although this might be due to the potential errors caused by the small sample size, previous studies also support our findings [19,32,33]. The accumulation of somatic aberrations in normal and atrophic endometrial glandular epithelium drives an evolving phenotype to EC [19,32]. Mota et al. revealed the presence of mutations in 50% (5/10) of the atypical hyperplasia cases and 3.70% (1/27) of the control patients (7 non-atypical hyperplasia, 7 leiomyomas, and 13 normal endometrium) [33]. Moreover, our study points to the following shortcomings of the current method and future directions. First, a large number of variants of uncertain/unknown significance (VUSs) lead to the failure of TPS to detect pathogenic mutations in all EC patients. In other words, pathogenic mutations cannot be separated from VUSs, although they were detected by TPS. Second, for targeted NGS panels, the variant allele frequency (VAF) limit of detection (LoD) ranges from 1% to 5%. However, the number of tumor cells in endometrial, cervical, and vaginal samples is relatively small, and the sensitivity of sequencing may not be sufficient to detect rare mutations (VAF < 1%). Even with unique molecular identifiers and a sequencing depth 30,000×, the SNV LoDs are about 0.1% [34,35]. Perhaps multiplex blocker displacement amplification, which can simultaneously provide good mutation sensitivity (LoD as low as 0.02%) and low cost, is an option [36]. Third, we did not collect control samples in potential risk groups, so we used HaplotypeCaller to call variants for non-matched NGS data. Although HaplotypeCaller has been used for somatic variant calling [37], it performs poorly for low-frequency somatic variants [38]. Collecting peripheral blood as a normal control in women with risk factors is recommended. Moreover, pathogenic mutations were found in ~28% of the women with risk factors. The gene mutation is an early but insufficient event in tumorigenesis [19]. According to mathematical models, developing a serous endometrial intraepithelial carcinoma from a *TP53* pathogenic variant takes decades [39]. Considering mutations may precede tumorigenesis, we have followed up with the potential risk group with pathogenic mutations to track whether these women have precursor lesions for EC in the future. Based on the follow-up results, we can ascertain whether mutations are potential biomarkers for assessing potential windows of intervention.

## 5. Conclusions

In summary, our results demonstrate that TPS is a sensitive method, and Pipelle endometrial biopsy is the optimal sample for detecting genetic lesions in EC. A small gene set consisting of *PIK3CA*, *PTEN*, *TP53*, *ARID1A*, *CTNNB1*, *KRAS*, and *MTOR* can achieve an excellent pathogenic mutation detection rate in Chinese EC patients and has the potential for large-scale application. For women with risk factors of EC, women with endometrial hyperplasia were more likely to have pathogenic mutations than those with benign lesions or unchanged endometrium.

## Figures and Tables

**Figure 1 cancers-15-03522-f001:**
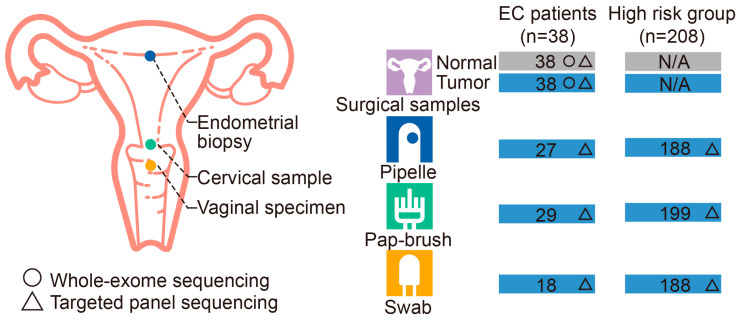
Schematic diagram of sample collection. The schematic drawing illustrates the sampling locations, methods, and numbers. Numbers in the gray box indicate the number of samples collected, and numbers in blue indicate the number of samples sequenced. The circle denotes whole-exome sequenced samples, and the triangle denotes targeted panel sequenced samples. WES, whole-exome sequencing; TPS, targeted panel sequencing.

**Figure 2 cancers-15-03522-f002:**
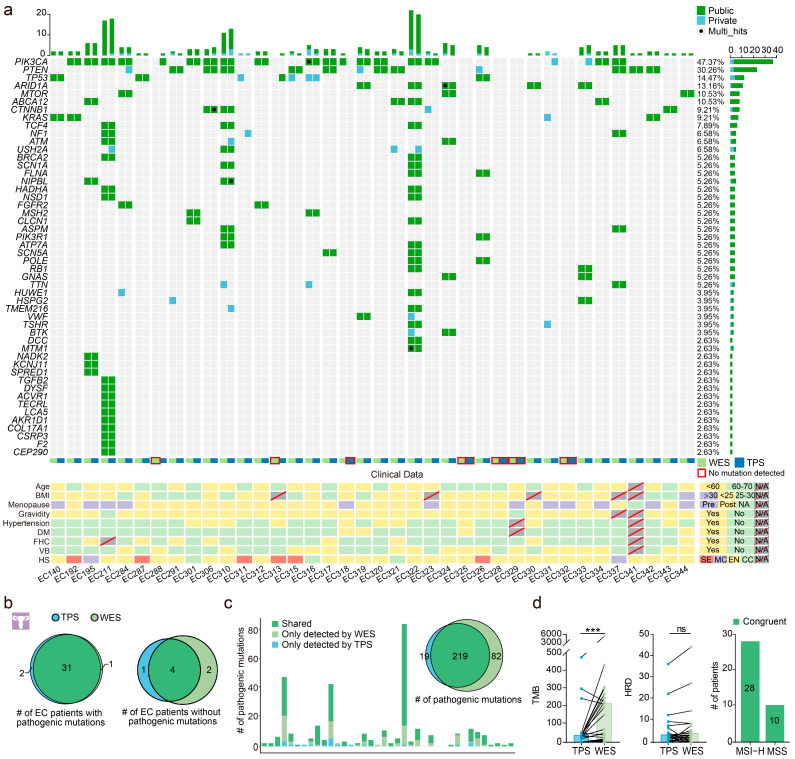
Performance of the custom panel in surgical specimens. (**a**) By-variant comparison of pathogenic mutations in the top 50 genes detected by TPS and/or WES in surgical specimens from endometrial cancer (EC) patients. Green represents mutations detected by both approaches, and blue represents mutations detected only by TPS or WES. The number of patients having a certain mutation is represented by the bar on the right. The number of mutations each patient has is represented by the bar at the top of the graph. The bottom heatmap shows the EC cohort’s clinical features and is color-coded according to the legends on the right. Every two columns of the mutation spectrum and each column of the heatmap represent one patient. The patient order of the mutation spectrum is consistent with the patient order of the heatmap. Slash lines indicate missing data. BMI, body mass index; DM, diabetes mellitus; FHC, family history of cancer; VB, vaginal bleeding; HS, histological subtype; SE, serous; MC, mixed carcinomas; EN, endometrioid; CC, clear cell. (**b**) The consistency of TPS and WES in detection of pathogenic mutations in EC patients. (**c**) Number of pathogenic mutations detected in the surgical tumor specimens of EC patients. The number of pathogenic mutations detected by both TPS and WES is indicated in dark green, only by TPS is indicated in grass green, and only by WES is indicated in blue. The Euler diagram shows the overlapping mutations detected by TPS and WES. (**d**) TPS and WES data consistency for TMB, HRD, and MSI analyses. ns, not significant; ***, *p* < 0.001 by Wilcoxon matched-pairs signed rank test.

**Figure 3 cancers-15-03522-f003:**
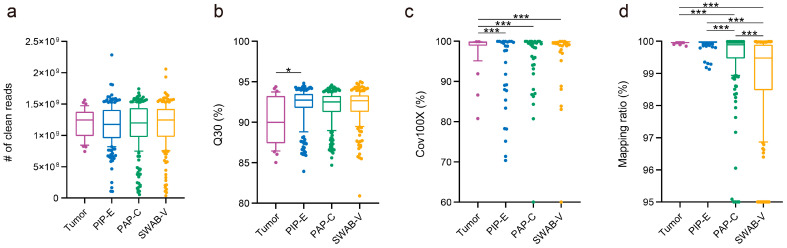
TPS data quality of endometrial, cervical, and vaginal samples. The number of clean reads (**a**), Q30 (**b**), Cov100× (**c**), and mapping ratio (**d**) of tumor, endometrial, cervical, and vaginal samples. *, FDR < 0.05; ***, FDR < 0.001 by Kruskal–Wallis test followed by Benjamini–Hochberg correction. PIP-E, Pipelle endometrial biopsy; PAP-C, cervical Pap-brush sample; SWAB-V, vaginal specimen.

**Figure 4 cancers-15-03522-f004:**
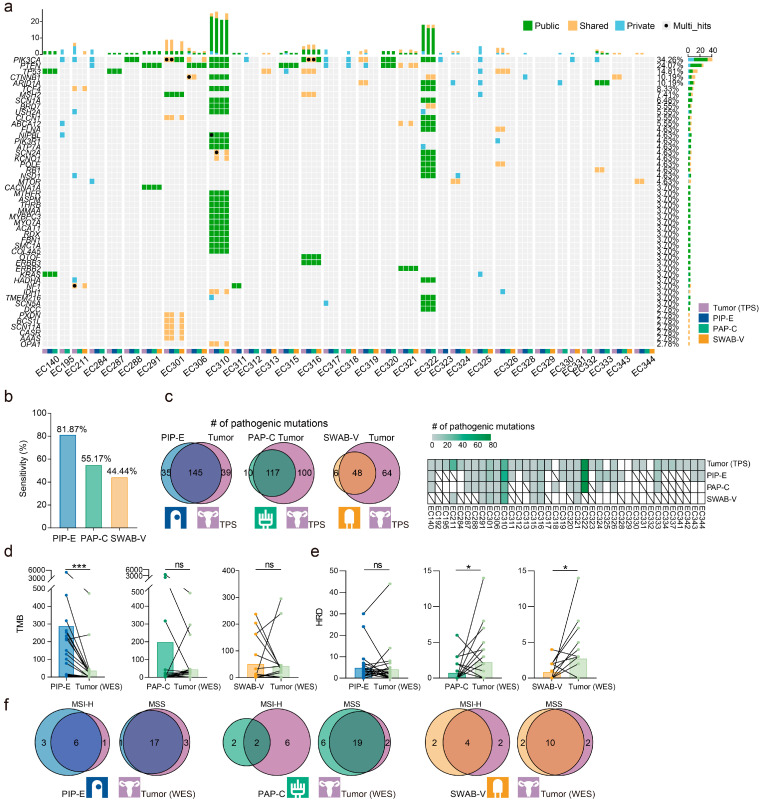
Diagnostic accuracy of endometrial, cervical, and vaginal samples for EC patients. (**a**) By-variant comparison of pathogenic mutations in the top 50 genes detected in EC patients’ endometrial, cervical, and/or vaginal samples. Public indicates mutations existing in all samples, shared indicates mutations existing in part of all samples, and private indicates mutations existing in one type of sample. PIP-E, Pipelle endometrial biopsy; PAP-C, cervical Pap-brush sample; SWAB-V, vaginal specimen. (**b**) The sensitivity in detection of mutations of EC of endometrial, cervical, and vaginal samples. (**c**) The number of overlapping mutations detected in endometrial biopsies and surgical tumor specimens, cervical samples and surgical tumor specimens, and vaginal specimens and surgical tumor specimens. The heatmap shows the number of pathogenic mutations detected in each sample. The darker the color, the greater the number of mutations. White boxes indicate no mutations detected. Slashes indicate that no sample was collected or the sample sequencing failed. (**d**–**f**) The consistency of TPS data between endometrial, cervical, vaginal, and surgical specimens for TMB, HRD, and MSI analyses. ns, not significant; *, FDR < 0.05; ***, FDR < 0.001 by Kruskal–Wallis test followed by Benjamini–Hochberg correction.

**Figure 5 cancers-15-03522-f005:**
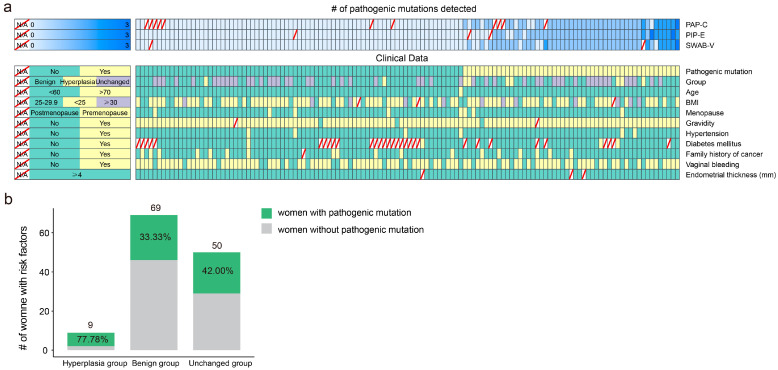
The number of pathogenic mutations and clinical features of women in hyperplasia, benign, and unchanged groups. (**a**) The heatmap on the top denotes the number of pathogenic mutations detected in three types of samples. The darker the color of the heatmap, the greater the number of mutations. Each column represents a patient. Slashes indicate that no sample was collected or the sample sequencing failed. The heatmap on the bottom denotes the clinical features of women in hyperplasia, benign, and unchanged groups and is color-coded according to the legends on the left. Each column represents a patient. Slash lines indicate missing data. (**b**) The pathogenic mutation detection rate of three groups. The number above the bar graph indicates the number of cases. Numbers inside the bar charts represent the percentage of women with pathogenic mutation(s). PIP-E, Pipelle endometrial biopsy; PAP-C, cervical Pap-brush sample; SWAB-V, vaginal specimen; BMI, body mass index.

**Table 1 cancers-15-03522-t001:** Baseline characteristics of the study population.

Variable	EC Group (*n* = 38)	Potential Risk Group (*n* = 208)	All (*n* = 246)
	*n* (%)	*n* (%)	*n* (%)
**Age at diagnosis, median (range)**	56 (33–65)	45 (23–75)	46 (23–75)
<60 years	22 (57.89)	199 (95.67)	221 (89.84)
60–70 years	15 (39.47)	6 (2.88)	21 (8.54)
>70 years	0 (0)	1 (0.48)	1 (0.41)
Missing data	1 (2.63)	2 (0.96)	3 (1.22)
**BMI (kg/m^2^)**			
<25	23 (60.53)	130 (62.50)	153 (62.20)
25–29.9	9 (23.68)	52 (25.00)	61 (24.80)
≥30	1 (2.63)	18 (8.65)	19 (7.72)
Missing data	5 (13.16)	8 (3.85)	13 (5.28)
**Histological subtype**			
Endometrioid	27 (71.05)	0 (0)	27 (10.98)
Serous	6 (15.79)	0 (0)	6 (2.44)
Clear cell	3 (7.89)	0 (0)	3 (1.22)
Mixed carcinomas	2 (5.26)	0 (0)	2 (0.81)
Atypical endometrial hyperplasia	0 (0)	1 (0.48)	1 (0.41)
Endometrial hyperplasia	0 (0)	8 (3.85)	8 (3.25)
Cystic endometrial atrophy	0 (0)	1 (0.48)	1 (0.41)
Endometrial polyp	0 (0)	4 (1.92)	4 (1.63)
Benign cystic glandular hyperplasia	0 (0)	41 (19.71)	41 (16.67)
Chronic endometritis	0 (0)	23 (11.06)	23 (9.35)
Proliferative phase	0 (0)	28 (13.46)	28 (11.38)
Secretory phase	0 (0)	22 (10.58)	22 (8.94)
Others	0 (0)	10 (4.81)	10 (4.07)
Missing data	0 (0)	70 (33.65)	70 (28.46)
**Menopausal state**			
Premenopausal	12 (31.58)	182 (77.12)	194 (78.86)
Postmenopausal	25 (65.79)	24 (10.17)	49 (19.92)
Missing data	1 (2.63)	2 (0.85)	3 (1.22)
**Vaginal bleeding**			
Yes	34 (89.47)	149 (71.63)	183 (74.39)
No	2 (5.26)	57 (27.40)	59 (23.98)
Missing data	2 (5.26)	2 (0.96)	4 (1.63)
**Endometrial thickness**			
<4 mm	0 (0)	3 (1.44)	3 (1.22)
≥4 mm	20 (52.63)	194 (93.27)	214 (86.99)
Missing data	18 (47.37)	11 (5.29)	29 (11.79)
**Gravidity**			
Yes	34 (89.47)	196 (94.23)	230 (93.50)
No	2 (5.26)	7 (3.3652)	9 (3.66)
Missing data	2 (5.26)	5 (2.40)	7 (2.84)
**Hypertension**			
Yes	10 (26.32)	16 (7.69)	26 (10.57)
No	26 (68.42)	190 (91.35)	216 (87.80)
Missing data	2 (5.26)	2 (0.96)	4 (1.63)
**Diabetes**			
Yes	4 (10.53)	9 (4.33)	13 (5.28)
No	32 (84.21)	138 (66.35)	170 (69.11)
Missing data	2 (5.26)	61 (29.33)	63 (25.61)
**Family history of cancer**			
Yes	8 (21.05)	25 (12.02)	33 (13.41)
No	28 (73.68)	179 (86.06)	207 (84.15)
Missing data	2 (5.26)	4 (1.92)	6 (2.44)

EC, Endometrial cancer; BMI, body mass index.

## Data Availability

The datasets generated and analyzed during the current study are available from the corresponding author upon reasonable request.

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
