# Peer review of "Diagnostic Potential of Endometrial Cancer DNA from Pipelle, Pap-Brush, and Swab Sampling"

_cancers, 2023, doi:10.3390/cancers15133522_

Round 1

Reviewer 1 Report

Materials and methods: I consider it incorrect to form a risk group based on the only one of the pre-set clinical indicators. I recommend that for the correct formation of the risk group, it is necessary to present classical approaches to the diagnosis of the pathological process nature in the endometrium (morphological analysis, ultrasound, CT diagnosis).

Results: I recommend selecting and identify the most adequate and informative mutations reflecting certain morphological changes in the endometrium.

Conclusions. I recommend, after making the mentioned remarks in other drafts of the manuscript, to transform the presented text and more correctly state the conclusions of the conducted research.

A minor stylistic revision is necessary.

Reviewer 2 Report

The article titled “Diagnostic Potential of Endometrial Cancer DNA from the 2 Pipelle, Pap-brush, and Swab Sampling" by  Yinan Wang et. al. is very important. Endometrial cancer is growing health concern worldwide. Early diagnosis of EC is critical for patient prognosis. 

 In this study, researchers aimed to find a safe and practical way of detecting early signs of endometrial cancer, which is crucial for effective treatment.

The study is correctly designed: a research question / problem was formulated, relevant research on a specific topic was detected, relevant research was critically selected and the results were evaluated and summarized.

This targeted panel sequencing method combined with ultra-deep sequencing is a promising tool for early detection of EC.

The authors indicate areas that should be further analyzed and in which it is worth to continue further experiments.

I suggest this paper to be published in Cancers

Author Response

Thank you for your positive feedback on our manuscript titled "Diagnostic Potential of Endometrial Cancer DNA from the Pipelle, Pap-brush, and Swab Sampling." We appreciate your recognition of the importance of early detection of endometrial cancer, and your acknowledgment of the design of our study.

We are pleased to hear that you found our targeted panel sequencing method combined with ultra-deep sequencing to be a promising tool for early detection of EC. We are grateful for your recommendation to publish our paper in Cancers and look forward to the opportunity to share our research with the wider scientific community.

Once again, thank you for your valuable feedback.

Reviewer 3 Report

In this study, Wang et al, aimed to find a safe and practical way of early detection of endometrial cancer. The researchers designed a panel targeting specific genes related to endometrial cancer, and tested it using samples from endometrial cancer patients and high risk women. The results showed that the panel produced high-quality data and detected genetic mutations with high sensitivity. The best sample type for detecting mutations was endometrial biopsy using Pipelle aspirator, which showed high consistency with surgical tumor specimens. These findings suggest that the custom panel combined with ultra-deep sequencing serves as a sensitive method for detecting genetic lesions from endometrial biopsy using Pipelle aspirator, which is a promising tool for early detection of endometrial cancer and could have significant clinical implications for patients. The research is properly designed, and the findings showed certain novelty and significance.

Author Response

Thank you for your positive comments on our manuscript titled "Diagnostic Potential of Endometrial Cancer DNA from the Pipelle, Pap-brush, and Swab Sampling." We appreciate your recognition of the design of our study and the novelty and significance of our findings.

Once again, thank you for your valuable feedback.

Round 2

Reviewer 1 Report

Indeed, the authors made a number of changes that contributed to a more correct presentation of the study results and made it possible to clearly identify the panel of genes associated with endometrial pathology. 

However, it is necessary to indicate the morphological changes in the endometrium of patients in the target group (with the presence of certain symptoms).

Since the formation of this group only on the basis of clinical characteristics without indicating the forms of the pathological process (glandular hyperplasia, complex and atypical hyperplasia) is incorrect. These additional data will allow the authors to adequately link the clinical symptoms and the obtained data on the frequency of pathological mutations in the panel of these genes.

The quality of the English language is satisfactory and in general allows you to understand the essence of the study.

Author Response

Thank you for your comment. We appreciate your feedback and agree that providing additional information on the morphological changes in the endometrium of patients in the target group would be beneficial. Thus, we have added more detailed pathological information about morphological changes in the endometrium for the potential risk group in Results section (line 195-198, line 316-323 and line 333-335), Table 1, Table S1, and Figure 5c. Of the 208 women with risk factors of EC, one women were detected to have atypical endometrial hyperplasia, eight had endometrial hyperplasia, one had cystic endometrial atrophy, 23 had chronic endometritis, four had endometrial polyp, one had adenomyosis, and 41 had benign cystic glandular hyperplasia. Moreover, 28 samples were identified as proliferative phase endometrium and 22 were identified as secretory phase endometrium. Pathogenic mutations were identified in women with atypical endometrial hyperplasia, cystic endometrial atrophy, and 75% endometrial hyperplasia, which were precancerous lesions of EC. However, pathogenic mutations were detected in less than 45% of women diagnosed with endometrial polyp (25%), benign cystic glandular hyperplasia (39.02%), chronic endometritis (26.09%), proliferative phase (42.86%) and secretory phase (40.91%). Pathogenic mutations occur more frequently in the precancerous lesions of endometrial cancer, although this might be due to potential errors caused by the small sample size. There are previous studies that can provide further evidence for our findings. Urick et al. mentioned in their review on molecular targets in EC that the accumulation of somatic aberrations in normal and atrophic endometrial glandular epithelium is crucial to tumorigenesis and progression of EC (Urick, M.E., Bell, D.W. Nature Reviews Cancer. 2019). Starting with a first mutation in normal glandular cells, it is the sequential accumulation of genetic damage within a continuous lineage that drives an evolving phenotype to endometrial cancer (Mutter et al. Cancer Res. 2014). Mota et al. revealed the presence of mutations in 5 of the 10 aspirates from atypical hyperplasia cases. By contrast, mutations were only identified in 1 of the 27 control patients (7 cases of non-atypical hyperplasia, 7 cases with leiomyomas and 13 with a normal endometrium) (Mota et al. Modern pathology. 2017).

Round 3

Reviewer 1 Report

According to the data supplemented by the authors, the group of patients at potential risk is heterogeneous in terms of morphological features. Therefore, I believe that they should be divided into at least 3 subgroups:

1 subgroup - hyperplasia, 2 subgroup - benign processes, 3 subgroup - unchanged endometrium in the phase of proliferation and secretion (I recommend using this group as a control). According to these three groups, sequencing results should also be presented.

The submitted manuscript is written in understandable English.
